# Recent Advances in Understanding the Complexity of Alcohol-Induced Pancreatic Dysfunction and Pancreatitis Development

**DOI:** 10.3390/biom10050669

**Published:** 2020-04-27

**Authors:** Karuna Rasineni, Mukund P. Srinivasan, Appakalai N. Balamurugan, Bhupendra S. Kaphalia, Shaogui Wang, Wen-Xing Ding, Stephen J. Pandol, Aurelia Lugea, Liz Simon, Patricia E. Molina, Peter Gao, Carol A. Casey, Natalia A. Osna, Kusum K. Kharbanda

**Affiliations:** 1Department of Internal Medicine, University of Nebraska Medical Center, Omaha, NE 68198, USA; ccasey@unmc.edu (C.A.C.); nosna@unmc.edu (N.A.O.); kkharbanda@unmc.edu (K.K.K.); 2Research Service, Veterans’ Affairs Nebraska-Western Iowa Health Care System, Omaha, NE 68105, USA; 3Department of Pathology, The University of Texas Medical Branch, Galveston, TX 77555-0419, USA; msprathi@utmb.edu (M.P.S.); bkaphali@utmb.edu (B.S.K.); 4Division of Pediatric General and Thoracic Surgery, Cincinnati Children’s Hospital Medical Center, Department of Surgery, University of Cincinnati, Cincinnati, OH 45229, USA; Bala.appakalai@cchmc.org; 5Department of Pharmacology, Toxicology and Therapeutics, University of Kansas Medical Center, Kansas City, MO 66160, USA; swang4@kumc.edu (S.W.); wxding@kumc.edu (W.-X.D.); 6Cedars-Sinai Medical Center, Los Angeles, CA 90048, USA; stephen.pandol@cshs.org (S.J.P.); aurelia.lugea@cshs.org (A.L.); 7Department of Physiology, Louisiana State University Health Sciences Center-New Orleans, New Orleans, LA 70112, USA; lsimo2@lsuhsc.edu (L.S.); pmolin@lsuhsc.edu (P.E.M.); 8Program Director, Division of Metabolism and Health Effects, National Institute on Alcohol Abuse and Alcoholism, Bethesda, MD 20892-6902, USA; gaozh@mail.nih.gov; 9Department of Biochemistry & Molecular Biology, University of Nebraska Medical Center, Omaha, NE 68198-5870, USA

**Keywords:** alcohol, fatty acid ethyl esters, pancreatitis, endoplasmic reticulum stress, autophagy, lysosome, transcription factor EB (TFEB)

## Abstract

Chronic excessive alcohol use is a well-recognized risk factor for pancreatic dysfunction and pancreatitis development. Evidence from in vivo and in vitro studies indicates that the detrimental effects of alcohol on the pancreas are from the direct toxic effects of metabolites and byproducts of ethanol metabolism such as reactive oxygen species. Pancreatic dysfunction and pancreatitis development are now increasingly thought to be multifactorial conditions, where alcohol, genetics, lifestyle, and infectious agents may determine the initiation and course of the disease. In this review, we first highlight the role of nonoxidative ethanol metabolism in the generation and accumulation of fatty acid ethyl esters (FAEEs) that cause multi-organellar dysfunction in the pancreas which ultimately leads to pancreatitis development. Further, we discuss how alcohol-mediated altered autophagy leads to the development of pancreatitis. We also provide insights into how alcohol interactions with other co-morbidities such as smoking or viral infections may negatively affect exocrine and endocrine pancreatic function. Finally, we present potential strategies to ameliorate organellar dysfunction which could attenuate pancreatic dysfunction and pancreatitis severity.

## 1. Introduction

Based on a 2018 report from the World Health Organization (WHO), alcohol abuse accounted for ~3 million deaths (5.3% of all deaths) worldwide in 2016 [1], and in the United States alone an estimated economic cost for excessive alcohol use was $249 billion in 2010 [2]. Extensive evidence indicates that alcohol abuse targets both exocrine [3,4] and endocrine functions [5] of the pancreas. There is supportive evidence that stresses and cellular abnormalities are the initiating factors for pancreatic dysfunction [6] and pancreatitis development [7,8].

Pancreatic islet cells are responsible for the secretion of metabolic hormones insulin and glucagon. Insulin is a critical hormone that regulates carbohydrate and lipid homeostasis. Several clinical and preclinical studies have revealed that alcohol consumption impairs basal and glucose-stimulated insulin secretion [6,9,10,11,12] and mechanistic insights have also been documented [5,6,11,13,14,15,16].

The pancreatic acinar cells are normally responsible for synthesizing, processing, and secreting huge quantities of digestive enzyme proteins that require folding and processing in the endoplasmic reticulum (ER) for appropriate transportation to the cell organelles for secretion upon physiologic stimulation. Studies have suggested that chronic alcohol consumption increases the mRNA levels of pancreatic digestive enzymes [17] but impairs their secretion [18], resulting in accumulation of zymogen granules in the gland. Alcohol-induced excessive unfolded/misfolded proteins form toxic aggregates in the ER of the pancreatic acinar cells causing unfolded protein response (UPR) and multiple organelle dysfunctions, which sensitize the acinar cell to premature activation of zymogens and activation of pro-inflammatory and cell death signaling [8,19,20,21]. Alcohol also induces the fragility of pancreatic zymogen granules [22], leading to increased intra-acinar activation of these digestive enzymes causing acinar cell death. The injured acinar cells induce endothelial cell dysfunction, immune cell infiltration, and activation of pro-fibrotic pancreatic stellate cells and wound healing responses. Thus, unresolved acinar and endothelial damage together with progressive profibrotic and pro-inflammatory responses lead to alcoholic pancreatitis. In addition, extensive acinar cell death results in pancreas atrophy [23,24] resulting in both exocrine and endocrine glandular dysfunction.

Chronic pancreatitis (CP) can result from recurrent episodes of acute pancreatitis (RAP) which results in permanent structural and functional damage and a high risk for pancreatic cancer development [25,26,27]. Although extensive epidemiological evidence indicates that alcohol abuse is a major risk factor for both RAP and CP [28,29,30], the volume, patterns, and timing of alcohol consumption required for the clinical development of pancreatitis are not clearly defined, and may vary depending on the presence of co-factors such as genetic mutations, smoking, and other lifestyle factors [31]. RAP or unexplained first episodes of acute pancreatitis (AP) in patients younger than 35 years of age are associated with pathogenic genetic variants in nearly half of the patients [32]. Smoking and alcohol consumption augment the effect of genetic mutations, stressing the potential for interactions between lifestyle factors and genetic susceptibility to pancreatitis [29,30,33,34,35].

Meta-analysis of observational studies have shown a dose-response relationship between average alcohol consumption and the risk of pancreatitis in men and women, and defined some thresholds required for increased risk of CP and RAP [28,31]. Other reports show that an estimated 10% of heavy alcohol users consuming 180 g/day for 10–15 years will eventually develop clinically overt CP [29,36]. A meta-analysis study of 51 international population-based reported studies [37], concluded that heavy alcohol use (>20 drinks per week on a regular basis) increases the risk of pancreatic diseases by nearly 40% compared to non-heavy, alcohol users; and the risk is modified by other co-factors including smoking, obesity, and diet habits. These reports support the concept that high average alcohol consumption increases the risk of pancreatitis, but more investigations are needed to determine how patterns of drinking, such as short-term and long-term heavy drinking as well as inter- and intra-individual variability affect the onset and clinical progression of AP to CP pathophysiology.

In this review we discuss the recent advances in understanding the complexity of alcohol-induced pancreatic dysfunction and development of alcoholic pancreatitis under the following pathophysiological themes:(i)Metabolic basis for alcoholic pancreatitis(ii)Role of the UPR in the development of alcoholic pancreatitis(iii)Role of impaired autophagy in the pathogenesis of alcoholic pancreatitis(iv)Chronic alcohol consumption dysregulates pancreatic endocrine function and exacerbates metabolic alterations in people living with human immunodeficiency virus (HIV)

## 2. Metabolic Basis for Alcoholic Pancreatitis

About 90% of the ingested ethanol is metabolized oxidatively in the liver by cytosolic alcohol dehydrogenase (ADH) and microsomal cytochrome P450 2E1 (Cyp2E1) to acetaldehyde, which is further metabolized to acetate by mitochondrial aldehyde dehydrogenase. Acetaldehyde is a reactive aldehyde and can alone cause oxidative stress in the target tissue. During chronic alcohol ingestion, ADH activity is commonly reduced in experimental animals and human subjects [38,39,40], and Cyp2E1-mediated ethanol metabolism results in the production of high levels of reactive oxygen species causing substantial oxidative stress.

The metabolic capacity to oxidize ethanol in the pancreas is much lower compared to the liver [41]. However, pancreatic acinar cells possess abundant amount of lipase(s)/esterase(s) that metabolize ethanol by nonoxidative pathways [3,42]. These enzymes are induced during chronic alcohol intoxication/abuse and catalyze the conjugation of alcohol with endogenous fatty acids to generate fatty acid ethyl esters (FAEEs) [3]. In fact, nonoxidative metabolism of ethanol to FAEEs is predominant in the pancreas and is several fold greater than that in the liver during chronic alcohol abuse/intoxication [3,42]. Further inhibition of hepatic ADH was reported to increase the formation of FAEEs in the pancreas [43]. However, FAEE formation appears largely dependent on a high body burden of alcohol. During chronic alcohol abuse, nonoxidative metabolism of ethanol to FAEEs by FAEE synthase seems to be one of the key pathways for its disposition in the pancreas. In addition to the pancreas, FAEEs have also been reported in the plasma, liver, adipose tissue, and heart of individuals diagnosed with a history of chronic alcohol abuse [3]. Of importance, FAEE synthase was reported to be higher in patients with alcohol-related pancreatitis [44]. Further, we found that plasma FAEE levels in humans positively correlate with blood alcohol levels [45]. In this study, a higher increase in total blood FAEE levels was observed in patients with a history of chronic alcohol ingestion compared to those with acute alcohol use, suggesting that reduced oxidative metabolism of ethanol may be associated with increased biosynthesis of FAEEs.

The relative role of oxidative and non-oxidative metabolites of ethanol during chronic alcohol abuse in inducing pancreatic damage is not well understood. It has been reported that the oxidative metabolism of ethanol sensitizes the pancreas and promotes cell damage in acinar cells [46,47]. In addition, ethanol and acetaldehyde alter a variety of signaling systems and cellular programs that elicit activation and fibroinflammatory responses in pancreatic stellate cells [48,49,50,51,52], the main cell type responsible for fibrosis in CP and pancreatic cancer [53,54].

The large amounts of highly lipophilic FAEEs that accumulate in the pancreas after acute and chronic ethanol exposure [25,42,55] have also been shown to cause toxicity to pancreatic acinar cells [36,43,56]. To understand the metabolic basis and the specific role of FAEEs in ethanol-induced pancreatic injury, hepatic ADH deficient (ADH^−^) deer mice were utilized [57]. ADH^−^ and hepatic ADH normal (ADH^+^) deer mice were fed daily for two months a liquid diet containing 1, 2 or 3.5 g% ethanol, doses relevant to those consumed by alcoholic subjects. A dose-dependent onset of lipid phenotype along with ~5-fold increases in FAEE levels in the pancreas were seen in ADH^−^ vs. ADH^+^ deer mice fed with 3.5 g% ethanol [25]. Further, FAEE levels were increased in pancreas and plasma of ADH^−^ deer mice but not in ADH^+^ mice when these mice were fed 3.5 g% ethanol for four months [57]. The increases in FAEE levels were associated with significant degenerative histological changes including pancreatic atrophy and acinar cell loss. Furthermore, ultrastructural analysis of the pancreas of ethanol-fed ADH^−^ deer mice revealed significant ER stress as evidenced by swelling and disintegration of ER cisternae in acinar cells, and activation of the protein kinase RNA-like endoplasmic reticulum kinase (PERK) branch of the UPR for ER stress [57].

The time and concentration-dependent formation of FAEEs and the cytotoxicity of ethanol exposure has also been demonstrated in rat pancreatic exocrine tumor (AR42J) cells [58] and in isolated human pancreatic acinar cells (unpublished data) exposed to ethanol. In these cellular systems, significant ER stress with upregulation of the ER regulator glucose regulated protein 78 (GRP78), and AMP-activated protein kinase deactivation was observed with increasing concentrations of ethanol. In addition, treatment of AR42J cells with ADH inhibitors significantly increased generation of FAEEs, cell death, and necrosis [59]. These findings underscore the key role of nonoxidative ethanol metabolism on the pathogenesis of CP.

## 3. Unfolded Protein Response in Pancreatitis Development

Acinar cells are specialized in the production, storage, and secretion of digestive enzymes and other proteins. To enable this high rate of protein production, acinar cells have extensive ER networks. The ER regulates the folding, processing, and trafficking of all secretory and membrane proteins, as well as the degradation of permanently misfolded/unfolded proteins by autophagy and the ubiquitin system. ER homeostasis is modulated by the UPR, a signaling program activated when the ER fails to properly process proteins, a condition termed ER stress [60,61]. The UPR aims to restore ER function and proteostasis, but severe or prolonged ER stress activates death programs. Two main branches of the UPR have been studied in pancreatitis: the inositol requiring enzyme 1 (IRE1) and the PERK branch. Activation of the IRE1 branch causes splicing of X-box binding protein 1 (XBP1u) mRNA, which gives rise to the potent transcription factor XBP1s (spliced form) that promotes adaptive responses to ER stress. XBP1s upregulates a large network of proteins needed to maintain ER function and the acinar cell secretory phenotype [62,63,64,65,66]. Activation of the PERK branch contributes to an adaptive UPR when well-tuned and transient, but persistent PERK activation upregulates DNA damage inducible transcript 3 (DDIT3, or CHOP), a transcription factor that regulates effectors of programmed cell death, autophagy, and inflammation [61,67,68,69,70].

Chronic alcohol feeding in rodents does not cause pancreatitis but increases levels of XBP1s in the pancreas, an effect likely due to ethanol-induced oxidative stress and with many potential consequences. XBP1s control transcription of many chaperones and oxidoreductases required for ER disulfide bond formation and folding of digestive enzymes, as well as components of protein degradation systems designed to degrade misfolded proteins. In acinar cells, XBP1s also have UPR independent functions and regulate post-ER protein trafficking and secretion, mitochondria-ER and lysosomal-ER communication, ER-dependent autophagy, and secretory vesicle formation [19,20]. Genetic inhibition of XBP1 in mice decreases digestive enzyme production and significantly reduces pancreatic secretion [19,20], while XBP1s overexpression in mice acinar cells increases pancreatic secretion [71]. To test the role of XBP1s in maintaining acinar cell homeostasis during alcohol feeding, its expression was genetically inhibited (Xbp1+/- mice). While partial inhibition had no effect in control-fed animals, it caused pancreatitis responses in the ethanol-fed animals. In ethanol-fed XBP1-deficient mice, widespread redox changes in ER proteins and ER dysfunction were observed and associated with histological evidence of pancreatitis [71,72]. All these observations suggest that the reason alcohol abuse does not cause pancreatitis in a large percentage of individuals is because of the ability of the pancreas to upregulate adaptation systems that prevent disease. This also led to the speculation that perhaps blocking the adaptation process could lead to pancreatitis development.

Due to epidemiologic observations that smoking augments the effects of alcohol abuse on pancreatitis [30,31], one hypothesis was entertained that smoking may promote alcoholic pancreatitis via inhibiting XBP1s. Indeed, a recent report indicated that although ethanol feeding upregulates XBP1s which prevents experimental pancreatitis, the addition of cigarette compounds or exposure to cigarette smoke results in inhibition of XBP1s and pathobiologic responses of pancreatitis [73]. In this study, smoking-induced XBP1s inhibition was associated with marked increases in proapoptotic CHOP and acinar cell death. Redox alterations are likely responsible for the observed effects of ethanol/smoking on XBP1s activity and CHOP levels as treatment with the anti-oxidant N-acetylcysteine (NAC) prevents the increase in CHOP and acinar death responses [73].

These studies indicate that, while alcohol alone can cause ER protein folding disorders, the adaptive UPR processes with alcohol abuse maintain ER homeostasis and prevent pancreatitis responses in the acinar cell. However, additional stresses such as smoking and possibly genetic mutations in digestive enzymes folded within the ER can incapacitate or overwhelm the adaptive and protective UPR resulting in a pathologic UPR and pancreatitis. Agents that attenuate ER stress are being tested in experimental models to determine potential benefits to treat pancreatitis in humans.

## 4. Impaired Autophagy-Lysosomal Pathway in the Pathogenesis of Pancreatitis

Autophagy is a cellular degradative process that requires the formation of double membrane autophagosomes that carry autophagic cargoes to lysosomes for degradation. Lysosomes sit at the end stage of the autophagic process and play a critical role in the completion of the autophagy process (or autophagic flux). Thus, it is critical to maintain the number and quality of lysosomes to meet the needs of autophagic degradation in response to various types of stresses. Lysosomal biogenesis is regulated by a family of proteins belonging to the microphthalmia family of basic helix-loop-helix-zipper (bHLH-Zip) transcription factors (MiT family) [74], which include transcription factor EB (TFEB), transcription factor E3 (TFE3), transcription factor EC (TFEC), and microphthalmia associated transcription factor (MiTF) that regulate the transcription of lysosomal biogenesis and autophagy [75]. These transcription factors bind to the conserved 10-base E-box-like sequence called coordinated lysosomal expression and regulation (CLEAR) motif [76]. Therefore, TFEB coordinates an efficient transcriptional program to control cellular degradation and facilitates intracellular clearance by regulating expression of genes for both lysosomes and autophagy. TFEB and its family of proteins are mainly regulated at the posttranslational level. TFEB is phosphorylated by MAPK1 (mitogen-activated protein kinase 1) and MTORC1 (mechanistic target of rapamycin complex 1) at Ser142, Ser211, and Ser122, resulting in the cytosolic retention and proteasomal degradation of TFEB [77,78,79]. In contrast, PRKCβ (protein kinase Cβ) phosphorylates TFEB at Ser461, Ser466, and Ser468, which stabilizes and activates TFEB [80].

Emerging evidence implicates an impaired autophagy-lysosomal pathway in the pathogenesis of pancreatitis [81,82,83]. Accumulation of autophagy vacuoles and positive LC3 puncta staining in acinar cells were described in a rat model of alcoholic AP elicited by alcohol feeding and inflammatory stimuli [84]. Decreased pancreatic lysosomal-associated membrane protein 1/2 (LAMP1/2) and lysosomal dysfunction have been reported in human and mouse alcoholic pancreatitis [85,86], which may account for the accumulation of large vacuoles in acinar cells in subjects with pancreatitis. Moreover, mice with genetic deletion of Atg5, an autophagy-related gene that regulates the formation of autophagosomes, or that of lysosomal-associated membrane protein 1/2 (LAMP1/2), led to the development of spontaneous pancreatitis [85,87].

It was recently demonstrated that in an experimental pancreatitis mouse model, cerulein decreased pancreatic TFEB proteins and TFEB-mediated lysosomal biogenesis, causing decreased lysosome numbers and insufficient autophagic flux [81]. Mice with a specific deletion of TFEB in acinar cells have exacerbated pancreatic damage induced by cerulein. Furthermore, mice with defective pancreatic lysosomal biogenesis by double deletion of TFE3 and acinar cell TFEB also develop spontaneous pancreatitis [81].

Using the chronic feeding plus binge (Gao-binge) alcohol model, two laboratories recently reported that alcohol increased levels of serum amylase and lipase [88,89]. Gao-binge alcohol also increased pancreatic edema, accumulation of zymogen granules, and expression of inflammatory cytokines. All these are hallmarks of pancreatitis although the extent of the pancreatitis was less severe compared with cerulein-induced pancreatitis in mice. Just as observed with cerulein treatment [81], mice subjected to a Gao-binge alcohol regimen also had decreased pancreatic TFEB and TFEB-mediated lysosomal biogenesis resulting in insufficient autophagy in mouse pancreas. Remarkably, cerulein treatment activated, whereas Gao-binge alcohol inhibited, mechanistic target of rapamycin (mTOR) in mouse acinar cells, suggesting alcohol inhibits TFEB in a mTOR-independent manner. Indeed, further investigations revealed that Gao-binge alcohol increased the levels of phosphorylated extracellular-signal-regulated kinase (ERK) in mouse pancreas, suggesting alcohol may inhibit acinar TFEB through MAPK-mediated phosphorylation of TFEB. How alcohol and cerulein activate different kinases to inactivate TFEB is currently not clear and needs to be studied in the future. Nevertheless, overexpression of TFEB in mouse pancreas attenuated Gao-binge alcohol-induced pancreatic injury. More importantly, human pancreatitis tissues either from alcohol or non-alcohol etiology show decreased TFEB nuclear staining. Together, these recent findings strongly indicate that impaired TFEB-mediated lysosomal biogenesis and autophagy play critical roles in promoting the pathogenesis of pancreatitis. Ongoing work is underway to investigate the mechanisms by which TFEB is inactivated during alcohol consumption, and hopefully to identify small molecules targeting TFEB for preventing/treating pancreatitis. Further evidence suggests that autophagy may help to selectively remove fragile zymogen granules and in turn prevent pancreatitis [90,91].

## 5. Chronic Alcohol Consumption Dysregulates Pancreatic Endocrine Function and Exacerbates Metabolic Alterations in People Living with HIV

In addition to the vast data on the risk for alcohol-mediated effects on pancreatitis development, there is increasing evidence that chronic alcohol consumption also impairs endocrine pancreatic function. Clinical [92,93,94] and preclinical studies [95,96,97] demonstrate that at risk alcohol use promotes metabolic dysregulation and is an independent risk factor for development of type 2 diabetes [98,99]. Type 2 diabetes is characterized by insensitivity to insulin action in liver, muscle, and adipose tissue progressing to impaired pancreatic insulin release and eventually β-cell death. While most studies have focused on how alcohol alters peripheral tissue insulin sensitivity (reviewed in [12]), little is known regarding the effect of alcohol on pancreatic endocrine function. The overall morbidity and mortality of subjects with acute pancreatitis is higher in individuals with type 2 diabetes [100]. Thus, understanding endocrine pancreatic dysfunction associated with chronic alcohol consumption is imperative for developing therapeutic strategies to ameliorate the burden of disease.

Clinical studies have shown that alcohol decreases circulating basal insulin secretion [9], and circulating insulin and c-peptide in response to a glucose load [10]. Rodent studies reveal that chronic alcohol administration significantly decreases circulating insulin levels [11] and glucose-stimulated insulin secretion from the pancreatic islets [12]. Studies have revealed that the alcohol-induced elevation of circulating ghrelin, a hormone mainly secreted from the stomach, inhibits insulin secretion from pancreatic β-cells [11,13]. In other experimental studies, rodents on a chronic alcohol diet show decreased pancreatic expression of glucokinase [5], glucose transporter-2 [5], and gamma-aminobutyric acid (GABA) receptors [14], all potential mechanisms for decreased insulin release. Moreover, in vitro alcohol exposure decreases glucose-stimulated insulin secretion from human [15] and rodent [11,14,16] islets and increases β-cell apoptosis [5,6]. All these studies indicate alcohol-mediated impairment of pancreatic β-cell function that can negatively impact glucose homeostasis and metabolic regulation.

The metabolic stress mediated by alcohol, including a chronic inflammatory milieu, oxidative stress, and mitochondrial dysfunction leads to islet cell dysfunction and could result in the decreased key transcriptional factors regulating islet cell function. Increases in blood glucose increase the transcription of the insulin gene [101]. The canonical pancreatic transcription factors, Pdx-1 (pancreatic and duodenal homeobox-1), NeuroD1 (neurogenic differentiation 1)/Beta2, and MafA (V-maf musculoaponeurotic fibrosarcoma oncogene homologue A) are critical for insulin gene transcription and release [101,102,103]. Evidence in the literature suggest that increases in inflammation and oxidative stress decrease the expression of these transcription factors [104,105,106]. Pancreatic function is also epigenetically regulated, and it has been demonstrated that alcohol alters the epigenomic profile in the muscle, brain, and liver [107,108,109,110,111,112,113,114,115]. It is of interest to understand how alcohol-mediated histone modifications or altered microRNAs regulate the expression of the canonical transcription factors and thus affect insulin expression and release. Furthermore, similar mechanisms responsible for the pathophysiology of alcoholic pancreatitis can also result in impaired pancreatic endocrine function.

The detrimental effects of chronic alcohol consumption on pancreatic function is enhanced under comorbid conditions such as viral infections. The prevalence of alcohol use disorder (AUD) in people living with HIV (PLWH) is higher than the prevalence of AUD in the general population [116,117]. Antiretroviral therapy (ART) has significantly reduced patient mortality, and HIV infection has emerged as a chronic disease, increasing the incidence of non-acquired immunodeficiency syndrome defined comorbidities including metabolic dysregulation, insulin resistance (IR), and diabetes.

Studies in SIV-infected macaques have shown that chronic binge alcohol (CBA) alters metabolic homeostasis [118,119,120]. Results from a frequently sampled intravenous glucose tolerance test (FSIVGTT) with a third-phase insulin infusion (modified minimal model; MINMOD) showed that CBA/SIV animals have decreased disposition index (DI), an indicator of IR, and markedly reduced insulin release and c-peptide levels (impaired endocrine pancreas response) following a glucose load [120]. These metabolic alterations were not associated with fasting hyperglycemia or hyperinsulinemia, suggesting they likely precede a phase of overt glucose intolerance developing with SIV disease progression. Preliminary data show that CBA decreases pancreatic expression of insulin, insulin:glucagon ratio, and NKX2.2 and PDX1. Data also demonstrate that CBA decreased circulating levels of high molecular weight (HMW) adiponectin in SIV-infected macaques [120]. Similarly, several studies indicate that alcohol decreases circulating adiponectin levels [121,122,123,124,125]. Adiponectin is an insulin sensitizing adipokine that stimulates exocytosis of insulin granules, increases insulin expression [126], and promotes β-cell survival [127]. Whether the alcohol-mediated decrease in adiponectin contributes to the observed pancreatic islet dysfunction has yet to be investigated.

Thus, increasing clinical and preclinical evidence points to alcohol-induced pancreatic endocrine dysfunction, increasing the risk for metabolic comorbidities. Further studies are warranted to understand the crosstalk between the pancreas and other organs such as the gut, adipose tissue, and skeletal muscle that could potentially contribute to pancreatic endocrine dysfunction. Whether lifestyle modifications including exercise or diet interventions can improve pancreatic function directly or indirectly by improving muscle and adipose health should be examined. Moreover, continuing to gain a better understanding of how alcohol-mediated alterations in epigenomic mechanisms impair pancreatic function, both exocrine and endocrine, is likely to identify targets for therapeutic interventions.

## 6. Conclusions

Overall, this review on the metabolic basis of ethanol-induced pancreatic cell injury provides valuable information for a better understanding of mechanisms of pancreatic insufficiency and pancreatitis pathogenesis such that appropriate therapeutic strategies could be developed. Figure 1 summarizes the take-home message of our current understandings of mechanisms of ethanol-induced pancreatic dysfunction: (i) inhibition of hepatic ADH during chronic alcohol abuse could be a key metabolic event resulting in an increased formation of FAEEs via nonoxidative metabolism in the pancreas that ultimately promotes the pathogenesis of CP; (ii) an adaptive UPR maintains ER homeostasis with alcohol abuse and prevents pancreatitis responses in the acinar cell. However, additional stresses such as smoking and possibly genetic mutations in digestive enzymes can incapacitate or overwhelm the adaptive and protective UPR resulting in a pathologic UPR and pancreatitis, reaffirming that lifestyle factors can cause ER protein folding disorders; (iii) impaired TFEB-mediated lysosomal biogenesis and autophagy play critical roles in promoting the pathogenesis of pancreatitis; and (iv) chronic alcohol consumption alters pancreatic endocrine function contributing to increasing metabolic comorbidities among people living with HIV.

Based on these considerations, it is predicted that agents that can reduce ER stress and improve autophagic mechanisms may be beneficial for preventing or treating pancreatitis. Further, lifestyle modifications including exercise or diet interventions could also improve both exocrine and endocrine pancreatic function.

## Figures and Tables

**Figure 1 biomolecules-10-00669-f001:**
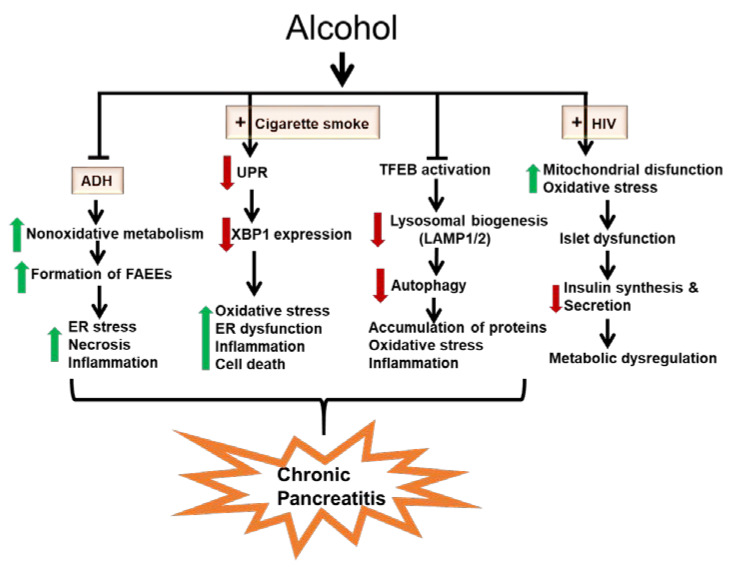
Schematic of ethanol-induced pancreatic dysfunction and pancreatitis development. Pathophysiological mechanisms involved in ethanol-induced pancreatic dysfunction include: (i) inhibition of hepatic alcohol dehydrogenase (ADH) during chronic alcohol abuse which could be a key metabolic event resulting in increased formation of fatty acid ethyl esters (FAEEs) via nonoxidative metabolism in the pancreas that ultimately promotes the pathogenesis of CP; (ii) an adaptive unfolded protein response (UPR) maintains endoplasmic reticulum (ER) homeostasis with alcohol abuse and prevents pancreatitis responses in the acinar cell. However, additional stresses such as smoking and possibly genetic mutations in digestive enzymes can incapacitate or overwhelm the adaptive and protective UPR resulting in a pathologic UPR and pancreatitis, reaffirming that lifestyle factors can cause ER protein folding disorders; (iii) impaired transcription factor EB (TFEB)-mediated lysosomal biogenesis and autophagy play critical roles in promoting the pathogenesis of pancreatitis; and (iv) alcohol can alter pancreatic endocrine function contributing to increasing metabolic comorbidities among people living with human immunodeficiency virus (HIV).

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
