# Peer review of "Recent Advances in Understanding the Complexity of Alcohol-Induced Pancreatic Dysfunction and Pancreatitis Development"

_biomolecules, 2020, doi:10.3390/biom10050669_

Round 1

Reviewer 1 Report

The authors comprehensively review biochemical aspects in the etiology of alcoholic pancreatitis.

The following revisions are necessary:

  • The English should be checked by a native speaker
  • Line 26: Please revise. I do not like the unscientific word „unhealthy“
  • Line 27 and throughout: in vivio and in vitor should be in italics
  • Line 33-37: completely revise. The text is completely uninformative. Instead of empty sentences such as “we provide insights …”, “we discuss …”, please state the actual  insights and results and conclusions of the review in an informative matter.
  • Line 52: annually in which country? Globally this should be much more? Perhaps data from WHO GISAH might be added.
  • Line 58 and throughout: please define all abbreviations at first instance (ER etc)
  • Line 154: revise, non-grammatical
  • Line 198: revise grammar
  • Line 208 and 211: stylistically I would avoid two instances of “interestingly”
  • Line 213: typo further
  • Line 230: AID?

Author Response

We thank you and the reviewers for considering our manuscript and for their helpful suggestions. We are resubmitting our manuscript for publication in the Biomolecules- special issue “Multi-Organ Alcohol-Related Damage: Mechanism and Treatment”. In this revised version, we have addressed all the comments of the reviewers as shown in this letter. All changes in the revised manuscript are underlined.

Reviewer comments

Reviewer: 1

The following revisions are necessary:

  1. The English should be checked by a native speaker

Response: This revised manuscript has been edited by a native English speaker.

  1. Line 26: Please revise. I do not like the unscientific word “unhealthy”.

Response: We have changed the “unhealthy alcohol use” to “excessive alcohol use” in the revised submission.

  1. Line 27 and throughout: in vivo and in vitro should be in italics

Response: We have italicized in vivo and in vitro throughout in the revised manuscript.

  1. Line 33-37: completely revise. The text is completely uninformative. Instead of empty sentences such as “we provide insights …”, “we discuss …”, please state the actual insights and results and conclusions of the review in an informative matter.

Response: We have revised the abstract and uninformative text has been changed according to the suggestions.

  1. Line 52: annually in which country? Globally this should be much more? Perhaps data from WHO GISAH might be added.

Response: We have added the 2018 WHO report on worldwide alcohol abuse in the introduction section of the revised manuscript.

  1. Line 58 and throughout: please define all abbreviations at first instance (ER etc)

Response: We have defined all the abbreviations when they appeared first time in the revised manuscript.

  1. Line 154: revise, non-grammatical

Response: We have revised the sentence for clarity.

  1. Line 198: revise grammar

Response: We have corrected the grammar.

  1. Line 208 and 211: stylistically I would avoid two instances of “interestingly”

Response: We have revised the sentences to avoid “interestingly”.

  1. Line 213: typo further

Response: We have corrected the typo.

  1. Line 230: AIDS?

Response: We have replaced the abbreviation “AIDS” with the full form “non-acquired immunodeficiency syndrome”.

Reviewer 2 Report

Where angels fear to tread.

This is an enthusiastic but incomplete overview of the current knowledge of the pathogenesis of alcoholic pancreatitis. Unfortunately, there are a substantial number of errors of fact and conceptualization implying a degree of amateurism in this field.

With respect to alcohol consumption and the risk of pancreatitis, there is a diversity in the literature about the level of alcohol consumption required for the clinical development of pancreatitis. Earlier work from Durbec and Sarles suggested that the risk was linear from zero consumption. Later studies have suggested a threshold (of 60 g or more of alcohol per day). Part of this diversity relates to differences in conceptualization; definitionally, many think that heavy alcohol consumption is a sine qua non for the diagnosis whereas those who emphasise co-factors such as smoking or genetic mutations are happy with the notion of lesser degrees of consumption which may be occurring coincidentally.

With respect to pancreatic alcohol metabolism, several authors have reported that ethanol oxidation is the dominant pathway (by far). This fact should not diminish the importance of the non-oxidative pathway/FAEE production as there is experimental evidence that FAEEs are produced in sufficient amounts in the pancreas to cause pancreatic organelle damage.

The evidence that ADH deficiency drives ethanol metabolism through the non-oxidative pathway has an experimental basis but not a clinical basis. There is no evidence that humans with alcoholic pancreatitis have ADH deficiency.

Chronic alcohol consumption by rodents DOES increase the pancreatic content of digestive and lysosomal enzymes and the fragility of zymogen granules and lysosomes thereby THEORETICALLY predisposing to intracellular contact between digestive and lysosomal enzymes thereby THEORETICALLY predisposing to autodigestion but the fact remains that autodigestion and pancreatitis are not observed in these rodent models of ethanol feeding.

The authors have virtually ignored the role of pancreatic stellate cells in the pathogenesis of chronic alcoholic pancreatitis. PSCs are the principal effector cells in pancreatic fibrosis. PSCs are activated by ethanol, an effect abrogated by antioxidants implicating the role of ethanol-induced oxidation.   

The role of smoking in the pathogenesis of alcoholic pancreatitis whether as an initiating risk factor or as an exacerbating factor needs much more attention from the authors.

The sections on UPR and autophagy can be shortened and made simpler for a more general audience. The major publication on the role of autophagy and alcoholic pancreatitis is by Forunato et al Ref 58.

To this reviewer, the section on HIV is incredibly speculative and tangential and has no place in this Review. Quite frankly, it looks like a paste from earlier work in a different context. This reviewer accepts that HIV positive individual may manifest overuse of alcohol but there is no evidence that HIV positivity is an independent risk factor for alcoholic pancreatitis.

Author Response

We thank you and the reviewers for considering our manuscript and for their helpful suggestions. We are resubmitting our manuscript for publication in the Biomolecules- special issue “Multi-Organ Alcohol-Related Damage: Mechanism and Treatment”. In this revised version, we have addressed all the comments of the reviewers as shown in this letter. All changes in the revised manuscript are underlined.

  1. This is an enthusiastic but incomplete overview of the current knowledge of the pathogenesis of alcoholic pancreatitis. Unfortunately, there are a substantial number of errors of fact and conceptualization implying a degree of amateurism in this field. With respect to alcohol consumption and the risk of pancreatitis, there is a diversity in the literature about the level of alcohol consumption required for the clinical development of pancreatitis. Earlier work from Durbec and Sarles suggested that the risk was linear from zero consumption. Later studies have suggested a threshold (of 60 g or more of alcohol per day). Part of this diversity relates to differences in conceptualization; definitionally, many think that heavy alcohol consumption is a sine qua non for the diagnosis whereas those who emphasise co-factors such as smoking or genetic mutations are happy with the notion of lesser degrees of consumption which may be occurring coincidentally.

Response: We have expanded this section in the revised manuscript to emphasize the diversity in the literature regarding the association of the volume and patterns of alcohol consumption with the increased risk of the clinical development of pancreatitis. We have added information on the role of cofactors (smoking/ genetic mutations) that increase this risk

  1. With respect to pancreatic alcohol metabolism, several authors have reported that ethanol oxidation is the dominant pathway (by far). This fact should not diminish the importance of the non-oxidative pathway/FAEE production as there is experimental evidence that FAEEs are produced in sufficient amounts in the pancreas to cause pancreatic organelle damage.

Response: We appreciate the reviewer for pointing out the significance of nonoxidative metabolism of alcohol (ethanol) and associated toxicity in the pancreas. It is well established that chronic alcohol consumption is the major cause of chronic pancreatitis. Since pancreas is main site for the formation of FAEEs, we hypothesized that chronic alcohol consumption increases the formation and accumulation of substantial amount of FAEEs in the pancreas causing lysosomal fragility. Lysosomal hydrolases released from fragile lysosomes can activate digestive zymogens causing acinar cell death and several other cellular stresses including ER stress and uncoupling of oxidative phosphorylation. We have reported a dose-dependent formation of FAEEs in the pancreas (~50-fold greater FAEE levels in the pancreas than that in the blood plasma) of hepatic ADH deficient deer mice fed 1, 2 and 3.5% ethanol daily for 3 months, and associated pancreatic injury (Kaphalia et al., 2010).

  1. The evidence that ADH deficiency drives ethanol metabolism through the non-oxidative pathway has an experimental basis but not a clinical basis. There is no evidence that humans with alcoholic pancreatitis have ADH deficiency.

Response: A direct relationship between chronic alcohol consumption and pancreatitis has been described in several clinical studies but reports regarding hepatic ADH deficiency and pancreatitis (liver-pancreatic axis) in alcoholic subjects are scarce. This liver-pancreatic axis in alcoholic pancreatitis stems from a common and well-documented observation that that chronic alcohol consumption reduces ADH activity in humans and experimental animals (Nuutinen Gastroenterol 21, 678, 1984; Sharkawi Life Sci 25, 2353, 1984; Panes Gastroenterol 97, 708, 1989; Panes Alcoholism: Clin Exp Res 17, 48, 1993; Kaphalia Alcohol 34, 151-158, 2004) and significantly increases the formation FAEEs and/or their toxicity in the pancreas (Laposata, Science 231, 497-499, 1986; Manautou, Toxicology 70, 303, 1991; Werner, Am J Physiol Gastrointest Liver Physiol 283:G65, 2002; Kaphalia Toxicol Appl Pharmacol, 246:154, 2010). Since metabolic basis as well as mechanism(s) of alcoholic pancreatitis are not well understood due to a lack of animal model, hepatic ADH deficient deer mice are being used to model alcoholic pancreatitis (Kaphalia Res Commun Chem Path Pharmacol 1, 173, 1996).

  1. Chronic alcohol consumption by rodents DOES increase the pancreatic content of digestive and lysosomal enzymes and the fragility of zymogen granules and lysosomes thereby THEORETICALLY predisposing to intracellular contact between digestive and lysosomal enzymes thereby THEORETICALLY predisposing to autodigestion but the fact remains that autodigestion and pancreatitis are not observed in these rodent models of ethanol feeding.

Response: Thanks for the comment. This reviewer is correct that chronic ethanol feeding alone seems to be not sufficient to induce obvious pancreatitis in rodent models. In our recent published paper in CMGH, we compared several ethanol feeding models in mice including chronic 4 weeks ethanol feeding, 10 days feeding without alcohol binge and 10 days feeding with acute binge (Gao-binge model). We found that only the Gao-binge alcohol model can lead to certain pancreatitis phenotypes such as edema and increased serum levels of amylase and lipase but not the chronic ethanol feeding only. In this study we found that TFEB, the key transcription factor for lysosomal biogenesis, was only impaired in the Gao-binge model but not in the chronic feeding only model. Another possibility to explain the different phenotypes among different alcohol models could be the changes of endotoxin. Alcohol binge may increase the gut leakage of endotoxin, which promotes pancreatitis as reported in reference Fortunato et al., 2009. We have now revised the manuscript to include the above points.

  1. The authors have virtually ignored the role of pancreatic stellate cells in the pathogenesis of chronic alcoholic pancreatitis. PSCs are the principal effector cells in pancreatic fibrosis. PSCs are activated by ethanol, an effect abrogated by antioxidants implicating the role of ethanol-induced oxidation.

Response: References were included regarding potential effects of ethanol metabolism on stellate cells.

  1. The role of smoking in the pathogenesis of alcoholic pancreatitis whether as an initiating risk factor or as an exacerbating factor needs much more attention from the authors.

Response: We have included references and studies indicating the role of smoking on pancreatitis.

  1. The sections on UPR and autophagy can be shortened and made simpler for a more general audience. The major publication on the role of autophagy and alcoholic pancreatitis is by Forunato et al.

Response: Thanks for the comment. However, we respectfully disagree with this comment as we thought the UPR and autophagy sections are important components of this manuscript and they were already very concise. Also, we have included a further discussion of Forunato’s paper (see Reference 76 in the revised manuscript).

  1. To this reviewer, the section on HIV is incredibly speculative and tangential and has no place in this Review. Quite frankly, it looks like a paste from earlier work in a different context. This reviewer accepts that HIV positive individual may manifest overuse of alcohol but there is no evidence that HIV positivity is an independent risk factor for alcoholic pancreatitis.

Response: This part of the review discusses the effects of alcohol on endocrine pancreas in the context of HIV/SIV. We have not discussed that HIV is a risk factor for alcoholic pancreatitis but do discuss that in the context of HIV/SIV, alcohol can exacerbate metabolic dysregulation by decreasing insulin secretion, an endocrine pancreatic function. The section also discusses some probable mechanisms of decreased endocrine pancreatic function, which can potentially contribute to the increased incidence of insulin resistance and diabetes seen in this population. This section of the review discusses pancreatic endocrine dysfunction, and the other sections discuss pathogenesis of alcoholic pancreatitis, overall indicating alcohol-mediated dysregulation of pancreatic function.  

Round 2

Reviewer 2 Report

The authors have made multiple changes to the text.

The section on HIV has been retained although this has nothing to do with the pathogenies of alcoholic pancreatitis and should be deleted.

Author Response

We thank the reviewers for their endorsement of our revised manuscript for publication. However, there was a minor concern expressed by Reviewer 2 who stated that “The section on HIV has been retained although this has nothing to do with the pathogenies of alcoholic pancreatitis and should be deleted.”

Response: We respectfully disagree with the reviewer since this manuscript, as the title reveals, is on the effect of alcohol consumption on the development of both pancreatic dysfunction and pancreatitis development by focusing on both the exocrine and endocrine function of the pancreas. Hence, while it is appropriate to include information on alcoholic pancreatitis, it is equally important to add in this review information on how chronic alcohol dysregulates pancreatic endocrine function and how a comorbid factor, such as viral infection, exacerbates metabolic alterations in people living with HIV.

In the submitted version, we have further elaborated on the effects of ethanol on endocrine pancreas and have then we specified that HIV as a cofactor for development endocrine dysfunction.

All our changes are underlined.